# TOWARDS ENVIRONMENTAL ROBUSTNESS IN DEEP REINFORCEMENT LEARNING

## ABSTRACT

Following the widespread application of Deep Reinforcement Learning (DRL) in robotics and other domains, adversarial attacks and robustness in DRL have also been widely studied in various threat models. However, most of them assume runtime access of the victim, which limits the feasibility of the attacks. To evaluate the robustness more practically, in this paper, we propose a threat model in which the attacker can only inflict static environmental perturbations on the initial state. By designing a preliminary non-targeted attack method and performing a case study on policy-based DRL agents, we show that the agents are still assailable in our threat model even though the capability of attackers has been severely limited due to the feasibility consideration. We also propose a defense framework, named **B**oosted **A**dversarial **T**raining (BAT), which incorporates a supervised kick-starting stage before adversarial training to avoid failure. Extensive experimental results demonstrate that our BAT framework can significantly enhance the robustness of agents in all situations while the existing robust reinforcement learning algorithms may not be suitable.

## 1 INTRODUCTION

The safety and robustness of Deep Reinforcement Learning (DRL) have been receiving increasing attention and have been studied in various domains, such as perturbation on the observation (Zhang et al., 2020; Oikarinen et al., 2021) or the action (Lee et al., 2020), data poisoning (Gunn et al., 2022; Panagiota et al., 2020), adversarial policies (Gleave et al., 2020), and multi-agent reinforcement learning (Lin et al., 2020; Guo et al., 2022).

Although effective, most of the existing threat models assume the strong capabilities of the adversary, resulting in limited practicability. For example, perturbing the observation or the action requires real-time write permission of the agent, and the poisoning attack needs access to the training data. Towards studying more practical attacks, we consider a new threat model where the adversary can not interfere with the agent but only tampers with the environment. As the example shown in Figure 1, we aim to confuse the agent by merely environmental perturbation, such as manipulating the positions of some irrelevant objects. Unlike adversarial policies (Pinto et al., 2017), environmental perturbation could be done beforehand instead of conducted by another agent at runtime.

Our threat model generally falls in the category of attacking the state space, whereas it is characterized by limiting the adversary in three aspects for better practicability. First, the state is resolved into the *environmental* state and the *agent* state, only the *environmental* state can be perturbed. Besides, the perturbations must be static and thus can only be exerted on the initial state. Finally, different from previous environment-related attacks (Schott et al., 2022; Chen et al., 2018), we restrict the perturbed state to be reachable from the standard initial state. The limitations ensure the alignment between our threat model and application scenarios in reality, where no runtime access to the target agent is required and the perturbation is promised to be realizable.

To measure the potential of environmental perturbations, we further design a white-box attack algorithm, which performs a non-targeted attack that impels the victim agent apart from its original actions to attack policy-based agents. The threat model and corresponding attack are validated in a cooking game, Overcooked (Carroll et al., 2019), where the agent controls two characters, and characters are allowed to put various objects on counters for coordination, which provides abundant

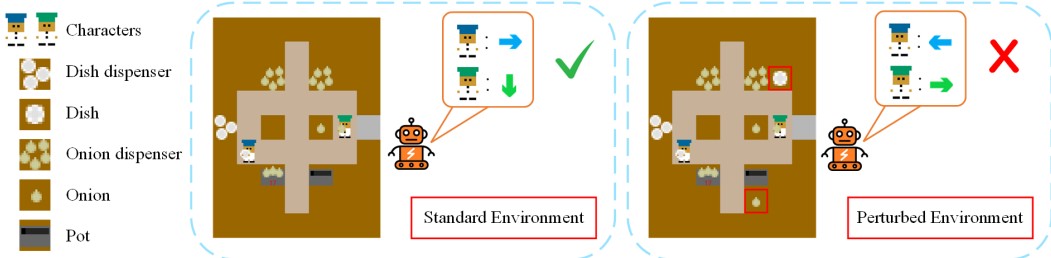

Figure 1: We show a representative case in a cooking game (Carroll et al., 2019), where the DRL agent can be incapacitated by deliberately placed objects as the environmental perturbation. The perturbations, which are marked in red square, are viable in the standard environment. A detailed introduction to the environment can be found in Section 4.

space for environmental perturbation. Experimental results show that our attack can significantly reduce the rewards of mainstream DRL agents, which substantiates that environmental perturbations on the initial states can incapacitate the agents trained in the vanilla environment. Even randomly inflicting perturbations on initial states can also evidently reduce the rewards. Such results reveal the vulnerability of DRL agents under environmental perturbation and indicate the existing room for improving the robustness.

While studying the attack problem to measure the risk, we also seek methods to defend and improve the existing DRL algorithms. As no existing robust reinforcement learning algorithm aims to defend the environmental perturbation, we propose a two-phase defense framework, which first kick-starts the agent via elaborated supervised learning and then fine-tunes the policy in the environment with generated adversarial initial states, namely **B**oosted **A**dversarial **T**raining (BAT). Comprehensive experiment results show that our BAT can greatly improve the robustness of DRL agents against environmental perturbation and even boost their performance in unperturbed environments while existing methods are not suitable in our setting.

The main contributions of this paper are:

- We propose a new threat model in which the saboteur can only perturb the initial environmental state in a reachable set regarding the environment and the standard initial state.
- We formulate the attack problem and design an effective attack method that performs non-targeted attacks on policy-based agents.
- We design the BAT framework which includes a supervised kick-starting stage and a fine-tuning stage to enhance the robustness of agents to environmental perturbations.
- We conduct comprehensive experiments for both attack and defense. The attack results show that the current DRL agents are vulnerable to environmental perturbations and their capabilities can be significantly decreased by our attack. The defense results show that our defense framework can effectively improve the resistance of the agents and outperform the baselines by a large margin.

In summary, our work establishes a foundation to study environmental perturbations and may enlighten future DRL research.

## 2 RELATED WORK

### 2.1 ADVERSARIAL ATTACKS IN DRL

As a hot research topic, the adversarial vulnerability of DRL has been widely discussed. Inspired by the success of adversarial examples in Deep Neural Networks (Szegedy et al., 2014; Su et al., 2019), researchers find that DRL is also vulnerable to little perturbations on inputs (Huang et al., 2017). Behzadan & Munir (2017) proposed a black box attack to Deep Q-networks (DQN) and validated the transferability of such an attack. By exploiting the characteristic of DRL, Lin et al. (2017) further proposed a strategically-timed attack to attack the agent at critical moments and an

enchanting attack to induce the agent to a certain state. Weng et al. (2020) proposed a model-based attack and extended the attacks into continuous domains. Universal adversarial perturbations that are static and agnostic to states are also studied to be powerful and efficient (Tekgul et al., 2022). Buddareddygari et al. (2022) noticed the importance of the physical availability of the attack and presented an algorithm to generate a static perturbation that can perform targeted attacks. Besides attacking the observations, Schott et al. (2022) successfully attacked the agents by disturbing the dynamics of the environment. Pan et al. (2022) studied various methods to improve the feasibility of attacks for application in reality and applied adversarial attacks to physical robots. Lin et al. (2020) performed adversarial attacks in cooperative multi-agent reinforcement learning (c-MARL) with a two-step attack that carefully deals with the characteristics of c-MARL. Similarly, Guo et al. (2022) comprehensively tested the robustness of c-MARL agents by attacking the state, action, and reward. Beyond defeating the agent, adversarial methods can also be used for evaluating the agents as a way to find the worst case (Uesato et al., 2019).

However, most existing attacks carried on the threat model that perturbs the observation of the victim agent and constrains the perturbation by the $L_p$-norm, which requires real-time access to the agent. Although previous works have shown the effectiveness of perturbing the environment (Schott et al., 2022) and the initial state (Panda & Vorobeychik, 2018), they do not aim to feasible environmental attacks. Starting from feasibility, our threat model restricts the environmental perturbations to be static and reachable in the original environment, which also makes the attack perceptually covert. Besides, the perturbations may also appear unintentionally due to the gap between the training environment and the production environment, thus the attacks can also be regarded as an evaluation of the robustness of agents.

## 2.2 ROBUST DRL

Since the vulnerability of DRL has become a wide concern, studying and improving the robustness of DRL has become another spotlight topic. Pinto et al. (2017) proposed RARL, which redefines the task as a zero-sum minimax problem and jointly trains a protagonist and an adversary. Similarly, training with an adversary embedded in environments has also been proven useful (Pattanaik et al., 2018). Methods that optimize the agent against adversarial perturbations at the training stage have also been shown effective, such as SA-MDP (Zhang et al., 2020), RADIAL (Oikarinen et al., 2021), and WocaR (Liang et al., 2022). Another perspective is detecting attacks and defending actively (Tekgul et al., 2022). Besides, Wu & Vorobeychik (2022) introduced an adversarial curriculum learning framework to boost the robustness of agents. Other than perturbations on observations, Schott et al. (2022) shows that training in environments with adversarial dynamics can also benefit the agent. RS-DQN (Fischer et al., 2019) utilizes distillation to train a student DQN along with a standard DQN thereby improving the robustness of the DQN. There are also works aiming to provide a guarantee or certification that the policy would not fail catastrophically when attacked by little perturbations (Lütjens et al., 2020; Everett et al., 2021). However, all the mentioned algorithms are designed under different threat models from ours, thus making it hard to resist our attack. To the best of our knowledge, no existing robust reinforcement learning algorithm is targeted to defend the environmental perturbations on the initial states.

## 3 METHODOLOGY

### 3.1 PRELIMINARIES

A typical Markov Decision Process (MDP) is defined as a tuple $< \mathcal{S}, \mathcal{A}, P, R, s_0 >$, where $S$ is a finite set of states, $\mathcal{A}$ is the finite set of available actions, $P : \mathcal{S} \times \mathcal{A} \times \mathcal{S} \mapsto [0, 1]$ is the transition probability, $R : \mathcal{S} \times \mathcal{A} \times \mathcal{S} \mapsto \mathbb{R}$ is a real-valued reward function, and $s_0 \in \mathcal{S}$ is the initial state. To concentrate on the environmental perturbations, we decouple the conventional state space $\mathcal{S}$ into $\mathcal{S} = \mathcal{S}^E \times \mathcal{S}^A$, where $\mathcal{S}^E$ stands for the environmental state space and $\mathcal{S}^A$ stands for the agent state space. Similarly, the initial state $s_0$ can be represented as $s_0 = (s_0^E, s_0^A)$, where $s_0^E \in \mathcal{S}^E$ and $s_0^A \in \mathcal{S}^A$ are the initial environmental state and the initial agent state, respectively.

As a case study, we concentrate on the policy-based DRL, which is one of the mainstreams. Specifically, each agent is associated with a policy $\pi$ that outputs a probability distribution over the action space given a state. Typically, we can note $\pi(a|s) \sim [0, 1]$ as the probability that the policy $\pi$ taking

action $a \in \mathcal{A}$ at state $s \in \mathcal{S}$. Alternatively, we use $\pi(s) \in \mathbb{R}^{|\mathcal{A}|}$ to denote the probabilities of all actions. For convenience, we also denote $\pi_{opt}(s) = \max_{a \in \mathcal{A}} \pi(a|s)$ as the action with the highest possibility that $\pi$ will choose. During the reinforcement learning process, a policy $\pi$ is trained to maximize the expected reward $\rho(\pi, s_0^E, s_0^A) = \rho(\pi, s_0) = \mathbb{E}_{\tau \sim P(\tau|\pi, s_0)} \sum_t R(s_t, a_t)$, where $\tau$ is the trajectory, $t$ stands for the time step, and $s_0$ is the initial state. For MDPs with infinite horizons, we can import a discount factor $\gamma$ to the objective to make it finite.

For attack, we aim to find out an initial environmental state $\widehat{s_0^E}$ that minimizes the expected reward $\rho(\pi, \widehat{s_0^E}, s_0^A)$ of a given policy $\pi$. The restriction is $D(s_0^E, \widehat{s_0^E}) \leq \epsilon$, in which $D$ is a function to measure the distance between states and $\epsilon$ is the threshold. It is noteworthy that $D$ may not be a conventional distance metric on the observation space such as $L_p$-norm, but a semantic metric that is specified by the experiment domain instead. Also, we would like to limit the adversarial initial environmental state to a feasible set $\widehat{\mathcal{S}^{\mathcal{E}}}$ to make the attack viable. In this paper, the feasible set is defined as all the possible environmental states that could be reached from the standard initial state $s_0$. Similarly, the defense problem can be defined as training a policy that is robust against the environmental perturbations on the initial state, which is a corresponding max-min problem.

However, the reward $\rho$ under an adversarial initial state is hard to estimate, simulating in the environment to get the reward is time-consuming and thus not applicable. Therefore, we design an attack algorithm in consideration of time efficiency. In this paper, we aim to attack and improve differentiable policies that generate a probability distribution over action spaces. By default, we assume the policies will follow the common practice in deep learning that first outputs a vector with $|\mathcal{A}|$ elements and then convert it to a probability distribution via the Softmax function. Nevertheless, our algorithms have the potential to be generalized to any agent.

## 3.2 ATTACKS

Similar to previous works, we generate adversarial examples against trajectories collected from the original environment, which keeps the computational efficiency while acquiescing to ignore the possible chain reaction caused by the perturbations. In practice, we observe that environmental perturbations would cause more capability generalization failure rather than goal misgeneralization introduced by Di Langosco et al. (2022). We also observe that out-of-distribution perturbations seldom interfere with the original trajectories of agents. From such observations, we further assume that environmental perturbations have a limited influence on optimal actions. Therefore, we aim to perform a non-targeted attack that induces the agent to deviate from its original actions. In this way, the objective of our attack algorithm is defined as:

$$\max_{\widehat{s_0^E} \in \widehat{\mathcal{S}^E}} \mathbb{E}_{\tau \sim P(\tau|\pi, s_0)}[\pi(a_t|s_t) - \pi(a_t|\hat{s}_t)], \tag{1}$$

$$s.t. D(s_0^E, \widehat{s_0^E}) \leq \epsilon, \ a_t = \pi_{opt}(s_t).$$

where $\tau$ denotes the trajectory, $\hat{s}_t$ is the estimated state after perturbations, and $\epsilon$ is the limitation of the distance. Stick to the insight that out-of-distribution perturbations incapacitate agents, we filter out the perturbations that are observed in regular trajectories over a frequency threshold $p_{freq}$. Considering the static nature of environmental perturbations, we assume the perturbations as time-invariant, which makes it simple to estimate the perturbed states $\hat{s}_t$ for $s_t$. Under such an assumption, we use $\hat{s}_t = \widehat{s_0^E} - s_0^E + s_t$ as a sketchy estimation. Although the assumption may not always hold, our method empirically shows enough ability to attack.

To further reduce the computational cost and make the algorithm scalable to Deep Neural Network-based policies with heavy computing, we take advantage of the differentiability of neural networks and approximately compute the significance of perturbation first-orderly to avoid tremendous calls of the policy network by:

$$\pi(a_t|s_t) - \pi(a_t|\hat{s}_t) \approx \frac{\partial \pi(a_t|s_t)}{\partial s_t} \Delta s, \tag{2}$$

where $\Delta s = s_t - \hat{s}_t = \widehat{s_0^E} - s_0^E$ is the estimated difference between the original state and the perturbed state, which is assuming invariant. Then, with collected trajectories $\tau$, we have a compu-

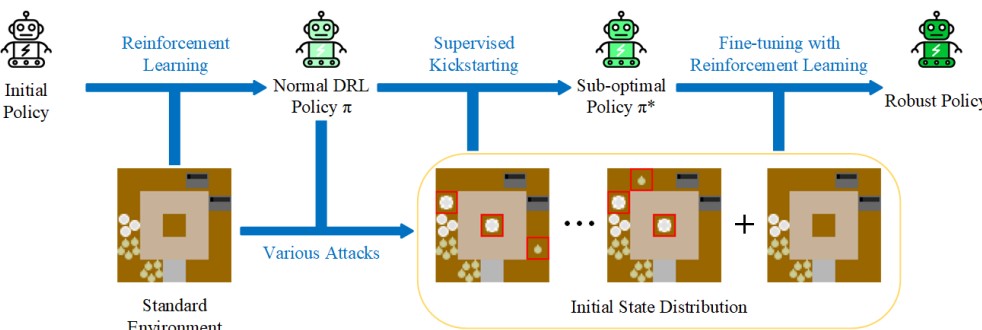

Figure 2: An illustration of the entire procedure of our BAT framework, which utilizes adversarial perturbations to adaptive defense.

tationally acceptable function to estimate the attack effectiveness given an adversarial state $\widehat{s_0^E}$:

$$J(\hat{s}_0^E) = \sum_{(s_t, a_t^*) \in \tau} \left[ \frac{\partial \pi(a_t^* | s_t)}{\partial s_t} (s_0^E - \hat{s}_0^E) \right]. \tag{3}$$

Where $a_t^* = \pi_{opt}(s_t)$ denotes the optimal action suggested by $\pi$.

The optimization of Equation (3) highly depends on the environment. For environments with discrete state space, gradient-based optimization may not be applicable. Generally, the problem can be solved via generic algorithms by taking the $J(\hat{s}_0^E)$ in Equation (3) as the fitness function and let $\mathcal{S}^E$ be the search space, while the solution may be further simplified regarding the characteristics of the environment as specified in Section 4.1.

## 3.3 DEFENSES

As the results will be discussed later in Section 4, the existing DRL agents expose their vulnerability under our attack algorithm. Therefore, we also seek methods to defend against attacks. A straightforward idea is the conventional adversarial training, *i.e.*, to fine-tune the DRL agents on the adversarial initial states. However, this may not practically work when the agents are severely disabled by the attacks, since the reinforcement learning process requires learning from trajectories with rewards, while the agent may fail to get any rewards in such cases and consequently fail the adversarial training. To solve such a problem, we introduce our BAT framework as shown in Figure 2, which first utilizes supervised learning to improve the capability of the agent in a teacher-forcing way as a kick-starting and then fine-tune the agent in the environment. The main insight is that the adversarial environmental perturbations have limited influence on the optimal behavior of the agent, which implies ignoring the disturbance may be still reasonable most of the time (although it is unlikely optimal).

In the first stage, our purpose is to prevent the agent from complete incapacity in the perturbed states with minimal effect on its capability. Here we conduct a case study on the actor-critic policy, which is a relatively complicated case and the method can be generalized to value-based or policy-based algorithms. Typically, in the case of actor-critic, there is a policy $\pi$ that computes the actions and a critic $V : \mathcal{S} \mapsto \mathbb{R}$ that estimates the value of states, while they may share some parameters. As a basic requirement, we hope the new policy $\pi^*$ can keep its capability in the pristine environment. From this aspect, we constrain the agent close to its original version by using the following loss functions:

$$\mathcal{L}_o = \mathbb{E}_{s \sim \tau} \left[ KL(\pi(s) || \pi^*(s)] + |V(s) - V^*(s)|, \tag{4}$$

where $KL$ stands for the Kullback–Leibler divergence, $V^*$ is the new value function.

Next, we hope the behaviors of the agent on perturbed states could be similar to those on original states as a decent start for fine-tuning. However, the outputs are not necessarily exactly the same, which may make the agent simply ignore the environmental changes. Intuitively, we expect the policy to act similarly as in the original states but with lower confidence. Inspired by the Knowledge

Distillation (Hinton et al., 2015; Papernot et al., 2016), we set a higher temperature $T$ for the Softmax function of the policy to acquire such labels for supervised learning. The policy with temperature $T$ is defined as:

$$\pi(s, T) = \left[ \frac{e^{z_i/T}}{\sum_{j=0}^{|\mathcal{A}|-1} e^{z_j/T}} \right]_{i=0,\ldots,|\mathcal{A}|-1}, \tag{5}$$

where $z \in \mathbb{R}^{|\mathcal{A}|}$ is the output of the last layer. The original policy is a special case where $T = 1$. Similarly, the values of disturbed states should be close to the original ones while not forced to be the same. Therefore we loosen the L1 loss of value function to achieve this purpose. The loss function on the disturbed data can be written as:

$$\mathcal{L}_p = \mathbb{E}_{s \sim \tau} \left[ KL(\pi(s, T) || \pi^*(\hat{s})) + max(|V(s) - V^*(\hat{s})| - \alpha|V(s)|, 0) \right], \tag{6}$$

where $\hat{s}$ denotes the estimated perturbed state, $\alpha$ is a hyper-parameter that controls the bound of proximity between the original values and disturbed ones. The loss terms above are then combined as $\mathcal{L} = \mathcal{L}_o + \beta \mathcal{L}_p$, where $\beta$ is a hyper-parameter that controls the weights of each part. In our practice, the perturbed initial states include adversarial initial states generated by the attack algorithm and random states in the feasible set. Different from the popular practice in knowledge distillation and student networks, our fine-tuning starts from the trained policy. It is worth emphasizing that the supervised learning phase does not immediately lead to a robust policy since it is designed to induce the policy to a good start of consequent fine-tuning, which will be done via regular reinforcement learning in the environment with a distribution of initial states $S_0$. Typically, the initial state distribution $S_0$ consists of the original initial state and the perturbed initial states used in kick-starting. It is noteworthy that although our defense method requires extra training, the fine-tuning process usually takes less time than training a standard DRL agent, thus our method does not bring excess computational burden.

## 4 EXPERIMENTS

### 4.1 TESTING ENVIRONMENT

Though environmental perturbations may be ubiquitous in the real world, it is hard to model in simulations or games. The mainstream benchmarks for RL may not explicitly incorporate the interaction between agents and the environment, *e.g.*, Mujoco (Todorov et al., 2012) and ProcGen (Cobbe et al., 2020). Therefore, we test DRL algorithms and all our methods in the Overcooked environment (Carroll et al., 2019), which has an interaction mechanism and was originally developed as a cooperative environment for studying multi-agent learning and human-AI collaboration. To fit the general DRL setting where there is only one agent, we use the environment in self-play mode, *i.e.*, the two characters are controlled by the same agent. Agents are trained to cook soup and serve it to certain locations. Typically, the agent needs to: put onions into a pot to make soup; dish out the soup once it is ready; and deliver it to a serving location. The action space of each character includes 6 actions: $wait$, $move\{up, down, left, right\}$, and $interact$. The agent can interact with environments, such as putting or taking objects to or from counters. Our experiments cover 6 various layouts as shown in the first line of Figure 3. The layouts have various difficulties so that the effectiveness of algorithms can be comprehensively tested.

Although Overcooked provides rich environmental states, our attack space is restricted by the standard initial states. Since all the counters are empty in the standard initial state, the removal of an object is not applicable in our experiments, leaving the following possible categories of unit perturbation: (1) Putting an onion on a reachable empty counter; (2) Putting a dish on a reachable empty counter; (3) Putting any onions in an empty pot. The perceptive distance between two states $D(s, \hat{s})$ is defined as the minimal number of unit perturbations to transform $s$ into $\hat{s}$.

The Overcooked simulator provides a lossless input in the space $\mathbb{R}^{26*w*h}$ corresponding to the state, where $w$ and $h$ are the width and height of the layout. Fortunately, each unit perturbation listed above exerts an independent influence on the observation. Let $P_1$ and $P_2$ be two compatible sets of unit perturbations, and $\Delta s_1$ and $\Delta s_2$ be their perturbations on the observation, $P_3 = P_1 \cup P_2$ will lead to permission $\Delta s_3 = \Delta s_1 + \Delta s_2$ on the observation. Therefore, instead of heuristic searching, we can simply select the unit perturbations with the highest negative impact and combine them.

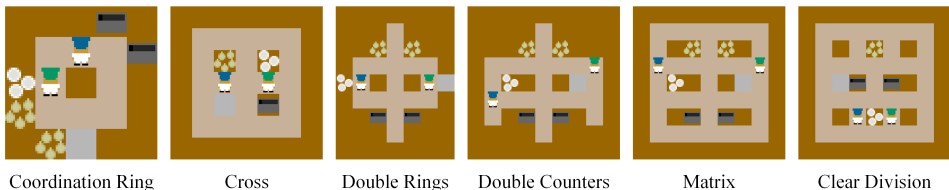

Figure 3: Visualization of the layouts used in experiments.

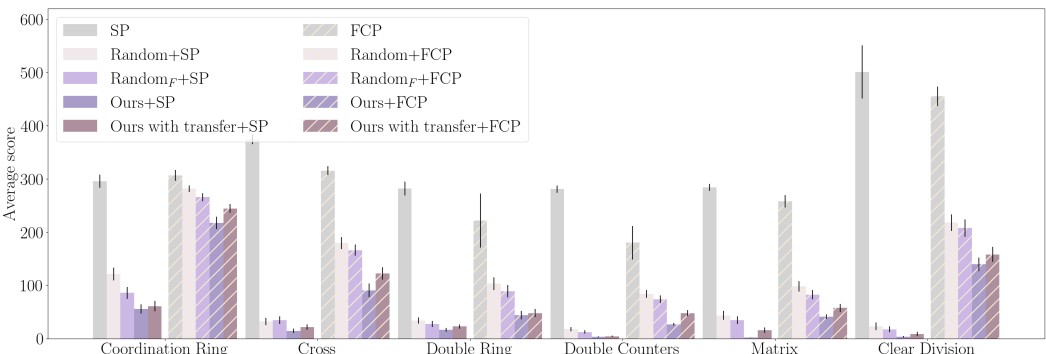

Figure 4: The mean scores and standard error across agents for SP and FCP.

## 4.2 ATTACK

### 4.2.1 EXPERIMENTAL SETUP

We conduct attack experiments on two popular DRL agents, Self-Play (SP) and Fictitious Co-play (FCP) (Strouse et al., 2021). In SP, the policy controls the two agents at the same time and is optimized with the joint trajectories. FCP is a strong algorithm designed for zero-shot coordination and human-AI collaboration, which trains the agent with a diversified group of pre-trained partners. For FCP, we train 4 partners with different seeds to form the partner pool and select 3 checkpoints with different levels of abilities for each partner. All agents are trained with the PPO algorithm (Schulman et al., 2017) for $1e7$ environment steps and have the same network architecture as in (Carroll et al., 2019). For both DRL algorithms, we train 5 independent agents.

To comprehensively evaluate our attack algorithm, we take the following methods into comparison as baselines:

- **Random**. The adversarial initial states are randomly selected from the feasible set.
- **Random$_F$**. Randomly selected adversarial initial states after filtering out the perturbations that have appeared in the collected trajectories over a certain frequency. The setting of $p_{freq}$ is the same as our attack algorithm.
- **Attack with transfer**. To evaluate the transferability of our attack, we test the performance of agents with perturbations generated by attacking other agents.

For all attacks, the perturbation limitation $\epsilon$ is set to 3. The number of outputs adversarial initial states $k$ of our attack algorithm is set to 10. To reduce the randomness, the $k$ of the random baselines was set to 40. For the evaluation of transferability, we use all the 40 adversarial initial states generated by attacking the other 4 agents. For each adversarial initial state, we run 100 games with 800 environment steps per game. We report the average score across all adversarial initial states and all 5 agents for each category. The evaluation settings remain the same throughout the experiment.

### 4.2.2 ATTACK RESULTS

The effectiveness of attacks on initial states can be revealed by the decrease in the average rewards. As the results are shown in Figure 4, our attack significantly reduces the rewards of both agents in

all layouts and clearly outperforms the random baselines. In some cases, the rewards even go down to near zero, such as SP in *Double Counters* and *Matrix*, indicating completely incapacitated agents. Such results demonstrate the effectiveness of our attack algorithm. Although the transferred attack shows less effectiveness than the white-box attack, it still consistently outperforms the random baselines, which indicates the decent transferability of our attack. We also notice that the $\text{Random}_F$ is slightly stronger than the Random, which somehow testifies to our insight that out-of-distribution states may incur failures of the agent. Furthermore, we find that even random perturbations can considerably decrease the rewards on all layouts, although they can be unintentional and are agnostic to the DRL agent. The result exposes that the mainstream DRL algorithms lack robustness to environmental perturbations. Even without a malicious adversary, the agent may fail due to an unexpected misalignment between the training environment and the testing environment.

Comparing the two DRL algorithms, the FCP agents show significantly better performance in resisting the attacks, although they may have lower scores in the vanilla environment. Recall that the FCP agents are trained with a set of diversified partners and have explored more states in their training data, this result fits the common perspective that the robustness of agents benefits from training with more diverse data. Moreover, when comparing across layouts, we find the attack is less effective in simpler layouts such as *Coordination Ring*, whereas the attack effect is much more significant in complex layouts. Such a phenomenon suggests that the success of attacks comes from unknown and unreasonable behaviors when the agents encounter out-of-distribution observations, and naturally leads to an insight that agents may benefit from training in more diversified environments and adversarial training.

## 4.3 DEFENSE

### 4.3.1 EXPERIMENTAL SETUP

For the adversarial initial states, we empirically select the top 5 outputs of our attack algorithm and sample 5 random initial states to form the set of perturbed initial states. The fine-tuning stage uses similar hyper-parameters but with only $8e6$ time steps, keeping the total computational cost of defense close to the original training process. More training details can be found in the supplementary. To comprehensively study the effectiveness of our proposed defense method, we employ several baselines for comparison, which are listed as follows:

- RADIAL(Oikarinen et al., 2021), a robust DRL framework that introduces an adversarial loss in optimization, which aims to resist adversarial perturbation restricted by the $L_p$-norm. Since no previous work targets environmental perturbations, we regard RADIAL as a representation of the existing robust DRL algorithm. We use RADIAL-PPO to train the self-play agents as a baseline. The perturbation bound $\epsilon$ is set to smoothly increase from 0 to $1/255$.

- Extra Training. To eliminate the influence of the difference in experimental settings between the original training and the fine-tuning, we additionally train the DRL agents for more time steps with the same setting as BAT without any defense method.

- Diversified Start. A natural and widely adopted way to improve the robustness of agents is to train from a distribution of initial states (Yang et al., 2022; Zhang et al., 2018), which is also applicable to defend against environmental perturbations. For this Baseline, the distribution of initial states is the same as the one used in BAT.

### 4.3.2 DEFENSE RESULTS

We present the mean scores of models under various attacks and corresponding standard errors in Table 1, where the best scores are in bold. Specifically, the scores achieved in the standard environments measure the capability of the agents, the random perturbations that have no pertinence to the agents measure their anti-interference performance, and the performance under attacks evaluates the resistance of agents. Generally, for both SP and FCP, our BAT can significantly improve their performance across all situations and preserve considerable capabilities under attack, which strongly demonstrates the effectiveness of our framework. It is surprising that BAT also significantly improves the performance even in the standard environment. Such a result indicates that the improvement of our framework comes from the improvement in the capabilities of agents, instead

Table 1: Quantitative Defense results (average scores with standard errors)

| Method | Attack | Coord. Ring | Cross | Doub. Rings | Doub. Coun. | Matrix | Clear Div. |
|---|---|---|---|---|---|---|---|
| Extra SP | No attack | $341.8 \pm 16.2$ | $399.5 \pm 11.6$ | $325.2 \pm 7.8$ | $304.7 \pm 7.3$ | $320.6 \pm 13.2$ | $549.5 \pm 41.1$ |
| | Random | $88.4 \pm 12.8$ | $36.1 \pm 8.2$ | $54.1 \pm 8.4$ | $29.8 \pm 6.6$ | $46.9 \pm 8.2$ | $23.5 \pm 6.2$ |
| | Our attack | $36.8 \pm 6.3$ | $5.0 \pm 1.2$ | $39.4 \pm 6.3$ | $9.3 \pm 2.6$ | $10.8 \pm 2.9$ | $4.9 \pm 1.0$ |
| Extra FCP | No attack | $365.6 \pm 4.2$ | $399.2 \pm 7.4$ | $327.6 \pm 14.6$ | $324.4 \pm 12.1$ | $307.6 \pm 8.0$ | $631.0 \pm 12.9$ |
| | Random | $234.7 \pm 13.9$ | $75.8 \pm 11.0$ | $44.5 \pm 6.9$ | $37.8 \pm 6.6$ | $16.8 \pm 4.8$ | $45.6 \pm 12.7$ |
| | Our attack | $121.3 \pm 12.9$ | $19.4 \pm 2.8$ | $37.7 \pm 9.6$ | $6.6 \pm 1.8$ | $1.0 \pm 0.3$ | $5.1 \pm 1.2$ |
| RADIAL | No attack | $273.6 \pm 9.6$ | $287.6 \pm 7.1$ | $204.6 \pm 6.7$ | $218.4 \pm 8.2$ | $202.2 \pm 8.7$ | $303.5 \pm 13.1$ |
| | Random | $138.8 \pm 10.1$ | $47.9 \pm 10.4$ | $43.9 \pm 7.3$ | $75.3 \pm 9.3$ | $40.4 \pm 7.5$ | $39.7 \pm 8.7$ |
| | Our attack | $103.1 \pm 9.1$ | $51.1 \pm 7.2$ | $15.1 \pm 2.8$ | $53.3 \pm 7.3$ | $14.7 \pm 3.7$ | $25.3 \pm 5.6$ |
| Div. Start | No attack | $311.7 \pm 20.4$ | $391.7 \pm 9.7$ | $304.1 \pm 23.0$ | $330.8 \pm 16.0$ | $330.9 \pm 8.4$ | $540.4 \pm 33.1$ |
| | Random | $240.2 \pm 8.9$ | $173.3 \pm 13.8$ | $176.2 \pm 13.1$ | $151.2 \pm 11.6$ | $172.8 \pm 11.2$ | $242.7 \pm 20.1$ |
| | Our attack | $221.1 \pm 11.2$ | $90.8 \pm 7.8$ | $153.7 \pm 14.0$ | $88.4 \pm 6.6$ | $106.1 \pm 8.2$ | $114.1 \pm 16.2$ |
| BAT+SP | No attack | $372.8 \pm 9.8$ | $459.7 \pm 14.3$ | $\mathbf{373.3 \pm 24.5}$ | $\mathbf{352.8 \pm 4.5}$ | $335.9 \pm 23.4$ | $612.2 \pm 35.6$ |
| | Random | $269.6 \pm 15.1$ | $279.4 \pm 14.7$ | $190.2 \pm 13.0$ | $180.6 \pm 13.8$ | $\mathbf{209.3 \pm 14.8}$ | $364.9 \pm 18.3$ |
| | Our attack | $197.2 \pm 10.9$ | $196.3 \pm 16.1$ | $138.0 \pm 14.7$ | $98.5 \pm 11.8$ | $\mathbf{107.6 \pm 12.3}$ | $247.8 \pm 15.3$ |
| BAT+FCP | No attack | $\mathbf{456.9 \pm 27.3}$ | $\mathbf{460.4 \pm 9.5}$ | $353.7 \pm 12.8$ | $340.9 \pm 19.0$ | $\mathbf{351.8 \pm 15.5}$ | $\mathbf{759.0 \pm 36.8}$ |
| | Random | $\mathbf{389.8 \pm 9.2}$ | $\mathbf{333.8 \pm 11.8}$ | $\mathbf{219.2 \pm 12.5}$ | $\mathbf{171.3 \pm 11.5}$ | $178.6 \pm 13.6$ | $\mathbf{467.7 \pm 24.2}$ |
| | Our attack | $\mathbf{308.6 \pm 12.3}$ | $\mathbf{206.3 \pm 10.1}$ | $\mathbf{178.9 \pm 12.3}$ | $\mathbf{120.7 \pm 13.0}$ | $58.3 \pm 9.3$ | $\mathbf{289.4 \pm 24.3}$ |

of increasing or decreasing the confidence of choices. We also notice that BAT + SP performs comparable to BAT + FCP, suggesting that our defense framework can produce diversified enough data for defending against environmental attacks while substantiating that our BAT remains effective in general tasks where no MARL algorithm is applicable.

Since the extra training is in self-play, it seems to bring the SP and FCP agents into convergence. The performance in unperturbed environments of FCP agents is significantly improved and is comparable to that of SP agents. Nevertheless, they are still vulnerable to environmental perturbations. In contrast, RADIAL shows somewhat resistance to environmental perturbations. However, it has significantly lower performance in standard environments, while the defense performance is still much weaker than our BAT. We deem that existing robustness algorithms designed to resist perturbations restricted by the $L_p$-norm are not suitable for defending against environmental perturbations. As the strongest baseline, training with diversified initial states also shows considerable defense performance, which is in line with our analysis that diversiform training data can improve the robustness of agents. Nevertheless, our BAT remains outperforming it, especially doing much better in the standard environment. Besides, BAT is an entire post-processing that has no limitation on the primary training, while training with diversified initial states may not be applicable for some algorithms, *e.g.*, algorithms including imitation learning (Carroll et al., 2019).

Another noteworthy point is that our attack is still significantly more effective than random perturbations. It confirms the ubiquitous power of our attack while implying that there is still room for improving the defense algorithms. We also present an ablation study of the necessity of each stage of BAT in the appendix.

## 5 CONCLUSION AND FUTURE WORK

In this paper, we propose a novel threat model in which the saboteur is only allowed to perturb the initial environmental state within feasible choices. We further design an algorithm to perform a non-targeted attack by generating adversarial initial states and propose the BAT framework to enhance the capabilities and robustness of agents. Extensive experiment results demonstrate that the mainstream DRL methods are vulnerable under our threat model and can be attacked by our algorithm. Furthermore, our defense method shows significant effectiveness in resisting attacks and even improves the capability of agents in the pristine environment and outstandingly outperforms the baselines including a representative existing robust reinforcement learning algorithm.

We preliminarily validate the existence of environmental vulnerability of DRL agents in the Overcooked domain. Although we make a few assumptions and simplifications regarding the 2D game nature of the Overcooked environment, our method is general and has the potential for further extension. Therefore, a natural direction of future work is to extend and study our proposed threat model and methods in more complex tasks, such as embodied intelligence or robots in the real world.

## REPRODUCIBILITY

We provide code and instructions for reproducing all our results in the supplementary materials. The modified version of the Overcooked environment is also included in the code.

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
