# OpenReview forum: "Towards Environmental Robustness in Deep Reinforcement Learning"
_ICLR.cc/2024/Conference — ICLR 2024 Conference Withdrawn Submission_

### Official Review · Reviewer_24xp · 2023-10-26

**Soundness:** 3 good
**Presentation:** 4 excellent
**Contribution:** 3 good
**Rating:** 6
**Confidence:** 4

**Summary:**

This paper proposes a method to achieve robust reinforcement learning policies that are robust to initial adversarial states. The authors first proposes a threat model that searches for adversarial initial states by formulating it as an optimization problem. In the second part, the authors proposed a two-stage defense method that warm starts a trained policy on a mixture of adversarial and random initial states via supervised fashion. This is done by using a loss function that constrains the policy close to the initial policy on the initial states, while constraining the policy to be close to the initial policy on adversarial states. The warm-started policy is then further fine-tuned via conventional RL training to achieve a robust policy. Experiments on an Overcooked environment shows that the proposed method is more robust.

**Strengths:**

The paper is mostly well-written and the main idea of the paper is relatively easy to follow along. The idea of adversarial attack on initial states is interesting and novel to the best of my knowledge. The defense framework proposed by the authors is also technically sound, although not necessarily novel as mentioned by the authors, since the idea of constraining policies is frequently used in RL. The results, in the given environment, are comprehensive and both the threat model and defense framework also shows significant improvement compared to the baseline.

**Weaknesses:**

My main concern is the generalizability and applicability of this framework (both for the threat model and subsequently the defense framework) to other environments and algorithms, as results were only shown for one environment, despite the comprehensive experiments.

Furthermore, certain parts of the main paper could use further clarification to improve the readability of the paper. Please refer to the next section on potential improvements.

**Questions:**

1. As mentioned above, my main question is about the generalizability and applicability of the threat model. In terms of generalizability, it would be more convincing if the authors could present at least some preliminary result on a separate environment. In terms of applicability, given that there's a lot of environments in which the states are non-image based. In that context, how applicable is the threat model, and it is not clear to me what is the notion of reachability in those environments. In such cases, will this threat model still be applicable? If not, the authors should also at least discuss the potential limitations of their approach.

2. Certain sentences may be too general. For example, "As no existing robust reinforcement learning algorithm aims to defend
the environmental perturbation, we propose a two-phase defense framework...". I believe there are existing literature which investigates such work. I suggest the authors revise such statements to be more specific to environmental perturbations in the form of initial states attacks

3. In section 3.1, what is the significance of decomposing $s_0$ into ($s^{E}_0$, $s^{A}_0$) ? In practice, how do we differentiate if an adversarial perturbation is on the environment or agent?

4. In section 3.2, the authors state "We also observe that out-of-distribution perturbations seldom interfere with the original trajectories of agents. From such observations, we further assume that environmental perturbations have a limited influence on optimal actions", and then "Stick to the insight that out-of-distribution perturbations incapacitate agents". These two statements seemed contradictory to me, could the authors clarify further?

5. I may have missed it but it is not clear to me how is Equation 3 optimized?

6. In Equations (4) and (5), is the $s$ sampled from the entire trajectory or are they just $s_0$?

7. In Section 4.2.1, "Attack with Transfer", are the attacks generated from attacking other agents of different seed or different algorithm?

8. In general, the main idea of the the warm-starting is to find a trained policy that is robust to both normal initial state and out-of-distribution initial states. How does this method compare to entropy-based methods such as SAC which also aims to find a good policy with high entropy?

From the appendix:
1. In Algorithm A line 7, the equation is denoted by $f$, while in the paper, the equation is denoted by $J$. Please keep it consistent
2. How are the parameters chosen in B.3? More importantly, how should a user know how long should the policy be kick-started before starting the fine-tuning? Is it empirical or based off some metric?

**Details Of Ethics Concerns:**

The attack methods presented could be replicated as one way to target RL-policies.

---

### Official Review · Reviewer_hrzc · 2023-10-29

**Soundness:** 2 fair
**Presentation:** 3 good
**Contribution:** 2 fair
**Rating:** 3
**Confidence:** 3

**Summary:**

This work focuses on the inflict static environmental perturbations on the initial state for DRL. This paper designs an attacking method to perturb the initial state of a trajectory and propose a defense framework, Boosted Adversarial Training (BAT) to defense the attack on the initial step of an environment.

**Strengths:**

- The writing is clear and easy to follow.
- This paper focuses on an interesting angle to tackle the adversarial robust RL problem: How to attack and defend over the attacks on the initial states.

**Weaknesses:**

1. Though the attacking angle looks new in this community, as the reviewers look into the problem formulation, some concerns raise. The difference between the environmental perturbation and model misspecification or other environment state perturbation works is not clearly distinguished. How does this work differ from environment state attack/non-robustness works (e.g. model misspecification but with misspecification only on the initial state)?
2. The discussions and the evaluations are all on the discrete benchmark. The attack on the initial environmental state is more like a perturbation problem. Why can’t the attacker just perturb all the allowed initial perturbation space? This work proposes a computationally effective way to reduce the perturbation cost, which limits the significance of the work.
3. This paper decouples the environmental states and agent states, which needs detailed explanation and justification. How is this decoupling done? How do you distinguish environmental states from agents?
4. To the reviewer, this paper’s proposal towards robustness is more inline with an abstract representation of the distance (vary from benchmarks to benchmarks) rather than the distance between absolute state value. Focusing on the robust domain seems mis-position this work.
5. Limited evaluation. There is only one discrete benchmark: Overlook, which significantly weakens the evaluation part.
6. The comparison with RADIAL does not seem to be a fair comparison to the reviewer. RADIAL only considers state observation perturbation, while this work changes the state itself for the initial step.

**Questions:**

- What’s an example of the agent state and environment state?
- How does this work compare with model misspecification papers? What if you assume the model misspecification is only for the first step?

---

### Official Review · Reviewer_HcCU · 2023-10-30

**Soundness:** 2 fair
**Presentation:** 3 good
**Contribution:** 2 fair
**Rating:** 5
**Confidence:** 4

**Summary:**

This paper introduces a new threat model where attackers can perturb the initial environmental state to evaluate the robustness of Deep Reinforcement Learning (DRL) agents. The authors propose a non-targeted attack method and a defense framework called Boosted Adversarial Training (BAT) to enhance DRL agent robustness against environmental perturbations. Experimental results demonstrate the vulnerability of current DRL agents to such perturbations and the effectiveness of BAT in improving agent resistance. The work contributes a novel perspective on environmental perturbations in DRL and has potential implications for future research in the field.

**Strengths:**

1. **Writing Quality and Structure:** The paper is well-written and structured, which is easy to follow the presented main ideas.

2. **Novelty of Environmental Perturbation:** The introduction of environmental perturbation through changes in initial states offers a novel perspective in the context of adversarial attacks and robustness in DRL.

3. **Threat Model and Defense Method:** The paper not only proposes a new threat model but also presents a corresponding defense framework, providing a comprehensive and coherent narrative.

**Weaknesses:**

1. Scalability: The paper primarily focuses on experimental settings within the Overcooked environment, and lacks experimentation in other, more typical environments. The extensive experimental assumptions and reliance on grid worlds may limit the generalizability and scalability of the proposed approach. This narrow scope raises concerns about the real-world applicability of the findings.

2. Excessive Assumptions: The paper introduces a multitude of assumptions that encompass the environment, agent, and perturbation aspects. These assumptions include the constraint that only the initial state can be perturbed, the assumption of white box, and the division of the state into environmental and agent states, searchable perturbations. While the authors could provide examples to clarify their motivations, the excessive number of assumptions significantly impacts the practical significance of this paper.

3. The related work section should include references to [1], which addresses the strongest adversarial attacks for DRL, and [2], which proposes diverse attack methods and a robust RL training method known as ATLA.

[1] Who Is the Strongest Enemy? Towards Optimal and Efficient Evasion Attacks in Deep RL. Yanchao Sun, Ruijie Zheng, Yongyuan Liang, Furong Huang. ICLR 2022.

[2] Robust Reinforcement Learning on State Observations with Learned Optimal Adversary. Huan Zhang, Hongge Chen, Duane Boning, Cho-Jui Hsieh. ICLR 2021.

**Questions:**

1. The defense framework, comprising a supervised kick-starting stage and a fine-tuning stage, appears promising. I am curious whether the authors have considered its applicability to counter other types of attacks, such as state perturbations. How versatile is this framework in addressing a broader range of adversarial challenges?

2. Long-Term Optimality of Environmental Attacks: When discussing attacks for DRL agents, it's important to consider their impact over the episode. Do environmental attacks have the potential to consider long-term optimality?

3. The paper mentions the attack budget \epsilon, which is a critical parameter in many adversarial settings. Could the authors elaborate on the methodology or criteria used to determine the appropriate value for \epsilon in their experiments?

---

### Official Review · Reviewer_Zxv9 · 2023-10-31

**Soundness:** 2 fair
**Presentation:** 3 good
**Contribution:** 2 fair
**Rating:** 3
**Confidence:** 4

**Summary:**

The paper proposes a new test-stage state perturbation attack against a reinforcement learning agent. The problem has been well studied recently, with various attacks and defenses proposed. While previous work typically considers the worst case where the attacker can perturb any perceived state, the paper considers the case where the attacker can only manipulate the initial state. Further, the paper assumes that the initial perturbation can go beyond l_p norm-based perturbations commonly considered in previous work by allowing it to change the semantics of the state. The paper proposes a new attack for initial state perturbation and a defense against it, which are evaluated using the Overcooked environment.

**Strengths:**

The proposed attack is allowed to modify the initial state only, making it easier to deploy in practice.

The strength of the proposed attack lies in the observation that the perturbation does not need to be constrained by an l_p norm as commonly assumed. Instead, one may consider semantics-aware attacks that produce feasible initial states beyond the l_p norm constraint. This observation is interesting and can potentially lead to new attacks and defenses being developed and a deeper understanding of robust RL against state perturbations.

**Weaknesses:**

One limitation of the paper is that the set of possible perturbations is assumed to be given and known to both the attacker and the defender. In particular, for the Overcooked environment used in the evaluation, the so-called unit perturbations are manually designed and are assumed to be known. In general, however, it is difficult to identify feasible perturbations that can change the semantics of states, which heavily rely on domain knowledge and can be challenging to extract automatically. Hence, it is unclear how to generalize the study to new environments from either attacker's or defender's perspective.

The attack proposed in Section 3.2 is far from being mature. In particular, the effect of the initial perturbation on future states, as shown in (2), is poorly estimated. Although the attack shows some effectiveness compared with random attacks in the Overcooked environment, a simple myopic attack that tries to modify the initial action distribution as much as possible without considering the whole trajectory might equally work.

The evaluation uses a single environment with a set of manually crafted perturbations, which is unsatisfactory. Further, the comparison with RADIAL is not convincing. First, the l_p perturbation budget is set to 1/255, which might be too small for the attack considered in the paper. Second, it is unfair to compare the vanilla RADIAL with BAT+FCP and BAT+SP since the former is not designed to solve the Overcooked game. It is more reasonable to integrate RADIAL and FCP or SP. In addition to RADIAL, the paper should also include results on SA-MDP (Zhang et al., 2020) and WocaR (Liang et al., 2022), two state-of-the-art defenses against state perturbations in RL, integrated with SP and FCP and adversarially trained using the attack given in the paper.

Semantics-aware state perturbations have been considered in Franzmeyer et al., 2023, which shows that a sophisticated perturbation attack can produce observations consistent with the state-transition function of the intact environment and is, therefore, hard to detect. The attack considered in the current paper can be viewed as a special case of the attack in Franzmeyer et al., 2023.

Tim Franzmeyer, Stephen Marcus McAleer, Joao F. Henriques, Jakob Nicolaus Foerster, Philip Torr, Adel Bibi, Christian Schroeder de Witt. Illusory Attacks: Detectability Matters in Adversarial Attacks on Sequential Decision-Makers. AdvML-Frontiers 2023.

**Questions:**

In the attack proposed in Section 3.2, it is unclear why considering the whole trajectory as in (3) is needed, given that the state estimate using (2) can be far from accurate. Would a simple myopic attack that manipulates the initial action distribution as much as possible work?

Why is supervised learning needed in addition to fine-tuning in training the robust RL policy? It would help to have an ablation study on this.